# Fracture and Fatigue of Dental Implants Fixtures and Abutments with a Novel Internal Connection Design: An In Vitro Pilot Study Comparing Three Different Dental Implant Systems

**DOI:** 10.3390/jfb13040239

**Published:** 2022-11-14

**Authors:** Sung-Woon On, Sang-Min Yi, In-Young Park, Soo-Hwan Byun, Byoung-Eun Yang

**Affiliations:** 1Division of Oral and Maxillofacial Surgery, Department of Dentistry, Dongtan Sacred Heart Hospital, Hallym University College of Medicine, Hwaseong 18450, Korea; 2Graduate School of Clinical Dentistry, Hallym University, Chuncheon 24252, Korea; 3Institute of Clinical Dentistry, Hallym University, Chuncheon 24252, Korea; 4Division of Oral and Maxillofacial Surgery, Hallym University Sacred Heart Hospital, Hallym University College of Medicine, Anyang 14066, Korea; 5Division of Orthodontics, Hallym University Sacred Heart Hospital, Hallym University College of Medicine, Anyang 14066, Korea

**Keywords:** Torx connection, BLX, TORX++, IU, dental implant, mechanical properties

## Abstract

The aim of this study was to compare the mechanical behaviors of three dental implant fixtures with different abutment connection designs. Three implant systems were studied: the control (BLX implant), test group 1 (TORX++ implant), and test group 2 (IU implant). Three samples from each group were subjected to static compression to fracture tests to determine the maximum fracture load, and twelve samples were exposed to fatigue tests that measured how many cycles the implants could endure before deformation or fracture. Detailed images of the implant–abutment assemblies were obtained using micro-computed tomography imaging, and fractured or deformed areas were observed using a scanning electron microscope (SEM). The mean maximum breaking loads of 578.45 ± 42.46 N, 793.26 ± 57.43 N, and 862.30 ± 74.25 N were obtained for the BLX, TORX++, and IU implants, respectively. All samples in the three groups withstood 5 × 10^6^ cycles at 50% of the nominal peak value, and different fracture points were observed. All abutment connection designs showed suitable mechanical properties for intraoral use. Microscopic differences in the fracture patterns may be due to the differences in the fixture design or abutment connection, and mechanical complications could be prevented by lowering the overload reaching the implant or preventing peri-implantitis.

## 1. Introduction

Dental implant restorations are a common treatment modality for replacing missing teeth [1,2,3]. To increase the success rate of dental implants, new fixture materials, surface treatments, and fixture and abutment designs are under constant development. Complications arising from the use of dental implants are unavoidable and are an important issue for both clinicians and patients. Among them, mechanical complications are mainly associated with the mechanical properties and connection designs of implant fixtures and abutments, including the loosening of screws or abutments, fracturing of veneer or ceramic crown, loss of retention, abutment sinking, and fracture of the fixture, abutment, or screw [4].

It is well-known that the mechanical stability of the fixture–abutment connection when withstanding masticatory force is essential for preventing the mechanical failure of dental implants [5,6]. Because the mechanical stability of this connection may be related to the connection design [7], continuous modifications and changes to the fixture–abutment connection designs are important. Recently, a new line of fixtures was launched using a titanium–zirconium (Ti–Zr) alloy with a Torx abutment connection design. According to the manufacturer, the novel Torx design is capable of high torque transmission due to its six contact points and 7° conical connection, which render it to attain higher mechanical stability and allow stress distribution. However, it is theoretically possible that radial stress may occur at the point of contact. To rapidly lower and dissipate the radial stress, an implant system with a modified Torx design for the abutment connection was developed.

This study aimed to compare the mechanical behaviors of three dental implant fixtures with different abutment connection designs. The null hypothesis was that no difference would be observed in the fracture load and fatigue limit due to the modification or difference in internal connection designs.

## 2. Materials and Methods

### 2.1. Types of Specimens and Preparation

A total of three implant systems were included in this study and were divided into three groups (Table 1): the control group, a 7° internal conical connection with a Torx design (BLX implant, Straumann, Basel, Switzerland); test group 1, a 6° internal conical connection with a Torx plus design (TORX++ implant, Korea Dental Implant Inc., Ansan, Republic of Korea); and test group 2, an 11° internal conical connection with a hex (IU implant, Warantec co., Ltd., Seongnam, Korea). All fixtures and abutment components in this study were fabricated from commercially pure titanium (grade 4) except for the BLX implant fixtures, which were made of Ti–Zr alloy. Each group comprised 15 specimens, three of which were subjected to static compression to fracture tests to determine the maximum fracture load [8] and twelve of which underwent fatigue tests to measure the number of cycles before the occurrence of deformation or fracture under cyclic loading. All fixture–abutment assemblies were connected according to the manufacturer’s instructions, and a torque value of 30–35 Ncm was applied as recommended. During this study, the test environment temperature and relative humidity were maintained at 24 °C and 42%.

### 2.2. Acquisition of Micro-Computed Tomography (μ-CT) Image

Before the cyclic loading test, the implant–abutment assemblies were investigated using a μ-CT imager (SkyScan 1273, Bruker, Kontich, Belgium) to obtain detailed fastening images. Samples were securely fixed onto Styrofoam, and 180° views of the samples were filmed in one frame at 0.3° intervals.

### 2.3. Static Compression to Fracture Test and Dynamic Cyclic Fatigue Test

The installation of specimens for all mechanical tests was carried out according to the ISO 14801:2016 protocol [9]. First, static uniaxial compression tests were performed to measure the maximum breaking load and strain to fracture of the material. Three implant-abutment specimens from each group were embedded into the resin to be exposed approximately 3.0 mm below the nominal bone level. A hemispherical cap was then placed onto the superstructure of the embedded specimen before fixing it to the fixation jig of the testing machine (Instron E3000, Instron, Norwood, MA, USA) so that the load direction formed an inclination of 30° to the specimen. Compressive load increasing at a rate of 1 mm/min was applied until failure occurred, and failure was defined as a fracture or permanent deformation of the implant–abutment assembly. The maximum compression to fracture load values of three specimens from each group were recorded, and their average maximum compressive load was set as the nominal peak value for the fatigue testing.

After the static compression to fracture test, fatigue testing was performed according to the ISO 14801:2016 recommendations to determine the number of cycles before fracture. The preparation and fixation of the specimens in the dynamic fatigue test were performed in the same manner as in the static compression test. Cyclic loading with a sine waveform was applied to each specimen with the ratio of the minimum load to the maximum load set to 0.1 at a frequency of 15 Hz. The reason for using a frequency of 15 Hz was that there was no significant difference compared with when a frequency of 2 Hz was used, and thus the test could be more quickly conducted [10]. The fatigue test was started with a load of approximately 90% of the average maximum compressive load obtained through the static load test, and the final fatigue limit, which was obtained by incrementally lowering the load, was recorded and defined as the maximum load under which the sample could withstand 5 × 10^6^ cycles [9]. The fractured or deformed specimens were then examined using a field emission scanning electron microscope (SEM) (INSPECT-50, FEI, Hillsboro, OR, USA). To investigate the area of failure and elucidate the fracture or deformation mechanism, the fractured or deformed area observed under a microscope was divided into three categories: fixture-level failure, abutment-level failure, and screw-level failure.

### 2.4. Statistical Analysis

The average and standard deviation of the maximum compressive load of each group and the mean value of the number of cycles in the fatigue test were calculated. All statistical analyses were performed using SPSS 21.0 software (SPSS Inc., Chicago, IL, USA).

## 3. Results

### 3.1. μ-CT Image Analysis

The detailed designs of the three implant systems were observed using frontal and axial μ-CT images (Figure 1). The BLX implant is the thinnest in the most coronal area of the fixture, and the thinnest part of the TORX++ implant is where the fixture and the abutment are in contact with the butt joint, two threads away from the coronal part; the IU implant is thinnest where the transition from the top edge of the fixture to the first thread occurs.

### 3.2. Result of Static Compression to Fracture Test

The BLX, TORX++, and IU implant samples showed mean maximum breaking loads of 578.45 ± 42.46 N, 793.26 ± 57.43 N, and 862.30 ± 74.25 N, respectively (Table 2). Load–displacement graphs for each group and among the three groups are shown in Figure 2 and Figure 3, respectively. When measuring the BLX samples’ maximum compressive load, the maximum load was measured past the yield strength value of the specimen. After applying the maximum value, the load value continued to decrease due to the deformation of the specimen, and fracture did not occur; thus, in the case of the BLX implant, the maximum load after the yield strength value of the specimen was regarded as the maximum breaking load.

### 3.3. Result of Dynamic Cyclic Fatigue Test

All samples in the three groups withstood 5 × 10^6^ cycles at 50% of the nominal peak value. All samples in the BLX group withstood up to 70% of the nominal peak value. As a result, the fatigue limits of the BLX, TORX++, and IU implants were 405 N, 397 N, and 431 N, respectively. The detailed results of the fatigue test are shown in Table 3.

### 3.4. Failure Modes on Microscopic Observation

Based on the microscopic examination using the SEM technique, the fracture area under cyclic loading for the fatigue test varied for each implant. In the case of the BLX implants, four out of five fractured specimens showed fixture-level fractures, with the remaining case showing a simultaneous fracture in both the abutment and fixture (Figure 4). A load of 80% of the nominal peak value led to fracture at the uppermost edge of the fixture, while a loading of 90% of the nominal peak value led to fracture at the level between the first and second threads or the beginning of the microthread.

For the TORX++ implants, all fractures occurred at the fixture level, and all were at the level of the thinnest point between the first and second thread (Figure 5).

Regarding IU implants, fractures of the abutment and fixture were mixed (Figure 6). Under a load of 70% of the nominal peak value, the abutment and fixture fractured together, and the fractured area of the fixture was the uppermost edge. Under 80% and 90% of the nominal peak value, the frequency of the abutment sole fracture increased. In the case of fixtures, fractures generally occurred at the uppermost edge, except for one sample in which the fracture occurred between the first and second threads.

## 4. Discussion

In the present study, static compression and cyclic fatigue tests were performed on implant systems with different abutment connection designs to compare the mechanical aspects. Considering the reported maximum loads of 108–299 N and 216–847 N in the anterior and posterior regions, respectively [11,12,13,14], the results indicate that all three implant systems can accommodate the normative requirements and show good mechanical properties. In addition, all three implants withstood up to 50% of the nominal peak value in the fatigue test; in the case of the BLX implant, it could withstand up to 70% of the nominal peak. Because the harshest environment is simulated as the worst-case scenario, it can be assumed that all three implant systems can favorably survive intraoral clinical conditions. Although statistical comparison among the three groups was impossible due to the small number of samples, it may be assumed that all three implants demonstrate good mechanical properties.

Static load and fatigue tests are considered the most accurate tests for generating clinically relevant data according to the principles and laws of mechanical engineering [15]. Therefore, static and fatigue tests in the present study were conducted in accordance with the regulations of ISO 14801:2016 [9]. Bernhard et al. [16] performed fatigue tests to compare implant systems with different implant fixtures and abutment configurations, and the authors found that Roxolid implants exhibited a 13–42% higher fatigue limit than cold-worked implants. In light of this, the TORX++ and IU implants, which showed fatigue limits of 397 N and 431 N, respectively, presented comparable results to the BLX implant, which showed a fatigue limit of 405 N. Therefore, both implant systems, with similar fatigue limits to Roxolid implants, demonstrate that improved fatigue limits can be expected to show good long-term results in the oral cavity. Furthermore, the fact that all three implant systems survived 5 × 10^6^ cycles at 50% of the nominal peak value supports the speculation of one previous study that suggested the use of a loading regimen of 40–50% of the nominal peak value for cyclic loading tests [8].

One unexpected result obtained in this study is the relatively low maximum compressive load of the BLX implant. The maximum compressive load of the BLX implant was 578.45 ± 42.46 N, which was lower than those of the TORX++ and IU implants, which were 793.26 ± 57.43 N and 862.30 ± 74.25 N, respectively (Table 2). Watanabe et al. [17] conducted experiments under the same conditions in an in vitro study for tapered conical connection implants and reported a static fracture load of 827.9 ± 14.3 N for a Roxolid implant connected with a titanium-based zirconia abutment. This is significantly different from the results obtained in this study, and it is presumed that the BLT implant was applied as a Roxolid implant in Watanabe et al.’s study [17], whereas in this study, the BLX implant was applied; the difference may be due to the fixture design or the composition of the abutment materials. In addition, as this study determined the maximum load beyond the yield strength of the specimen as the maximum compressive load, this difference may be due to the use of different measurement regimens in Watanabe et al.’s study [17], wherein loading was ceased when a displacement of 1 mm occurred. Therefore, additional studies with more specimens are needed to more accurately elucidate the maximum compressive load of the BLX implant system.

The three implant systems showed different aspects of failure modes that were observed microscopically under a cyclic load as the fatigue test. In the case of the TORX++ implant, all specimen fixtures were fractured between the first and second threads, i.e., the thinnest part of the fixture, regardless of the load applied (Figure 4). However, the BLX and IU implants did not show the fracture tendency seen in TORX++ implants. The BLX implants mainly showed fractures at the uppermost edge of the fixture, and as the load increased, the fractured areas gradually moved to the apex of the fixture (Figure 5). In the case of the IU implant, the frequency of fracture of the abutment alone increased as the loading increased, and the fractured area showed a tendency to move from the uppermost edge to the apex of fracture of the fixture, as in the fixture of the BLX implant (Figure 6). In light of the assumption of a 3 mm resorption of the marginal alveolar bone in ISO 14801, the fracture risk of the TORX++ implant increases when the bone level extends down to the thinnest point (between the first and second thread) of the fixture as the alveolar bone is resorbed. The BLX and IU implants have a high risk of fracture at the uppermost edge of the fixture; however, there is a tendency for the fractured areas in BLX and IU fixtures to descend downwards as the overload increases. Although there are no fatigue test results assuming that the alveolar bone is not resorbed in this study, the TORX++ implant can avoid fixture fracture by preventing the overload and peri-implantitis that leads to marginal bone resorption.

In contrast, fracture mainly occurs at the uppermost edge of the fixture in the BLX and IU implants, meaning that efforts should be made to avoid stress at the top edge of the fixture. In addition, in the case of the IU implants, it is considered that additional fracture evaluations of the abutment are continuously required. However, to establish a more precise evaluation, it is necessary to perform additional research through experiments in a situation where there is no marginal bone loss.

The present study has originality in the following respects. First, this is the first study to evaluate the mechanical properties of an implant system with the Torx plus design. To compensate for the disadvantages of the Torx design that may theoretically cause radial stress at the point of contact, the Torx plus design, in which the radial stress is lowered and rapidly dissipated, was developed, prompting us to perform a comparative study. Although statistically significant comparisons were not made, a continuous evaluation of the Torx plus design could be performed based on the results of this study. Second, this study is the first to analyze and present the fracture pattern of the BLX implants using the SEM technique after a fatigue test. The BLX implant is a Roxolid implant with a variable thread design, which was developed to obtain improved strength and primary stability. Its use is thus expected to increase in clinical situations, such as immediate implant placement after tooth extraction or the installation of narrow implants in the future. Therefore, the results of static and fatigue tests and the SEM analysis of fracture patterns in the BLX implants in this study can be considered meaningful.

The limitations of this study include the absence of statistical comparison due to the relatively small sample size. Although the study by Choi et al. [8] and the ISO 14801 protocol used a comparable sample size, the results may be ambiguous, thereby calling for further studies wherein a larger sample size is required. Furthermore, there is a possibility of mixed results due to differences in the implant fixture materials, abutment connection design, and taper angle of the internal conical connection among groups, which needs to be investigated in detail. In addition, fractography analysis or analytical calculations for the precise evaluation of fracture patterns could not be performed. Moreover, since this study only considered the requirements of ISO 14801 as the minimum material standards for dental implants, more complex conditions must be used for the study to gain more information and ensure the scientific robustness of the materials. Therefore, further studies employing more sophisticated evaluation methods other than the ISO protocol while using larger sample sizes under the same conditions are needed.

## 5. Conclusions

In conclusion, all implant systems with three different abutment connection designs in this study showed mechanical properties that are suitable for intraoral use. Although the fracture patterns of fixtures and abutments varied, which could be attributed to differences in the design of the fixture and abutment connection, including taper angle, were microscopically different. These mechanical complications could be prevented by lowering the overload on the implant or preventing peri-implantitis. Therefore, clinicians should carefully consider these characteristics of the implant systems.

## Figures and Tables

**Figure 1 jfb-13-00239-f001:**
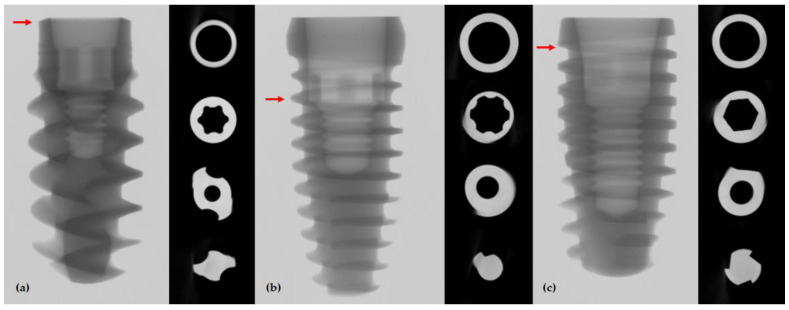
Frontal and axial μ-CT views: (**a**) BLX implant; (**b**) TORX++ implant; (**c**) IU implant. Red arrows indicate the thinnest part of the implant fixture.

**Figure 2 jfb-13-00239-f002:**
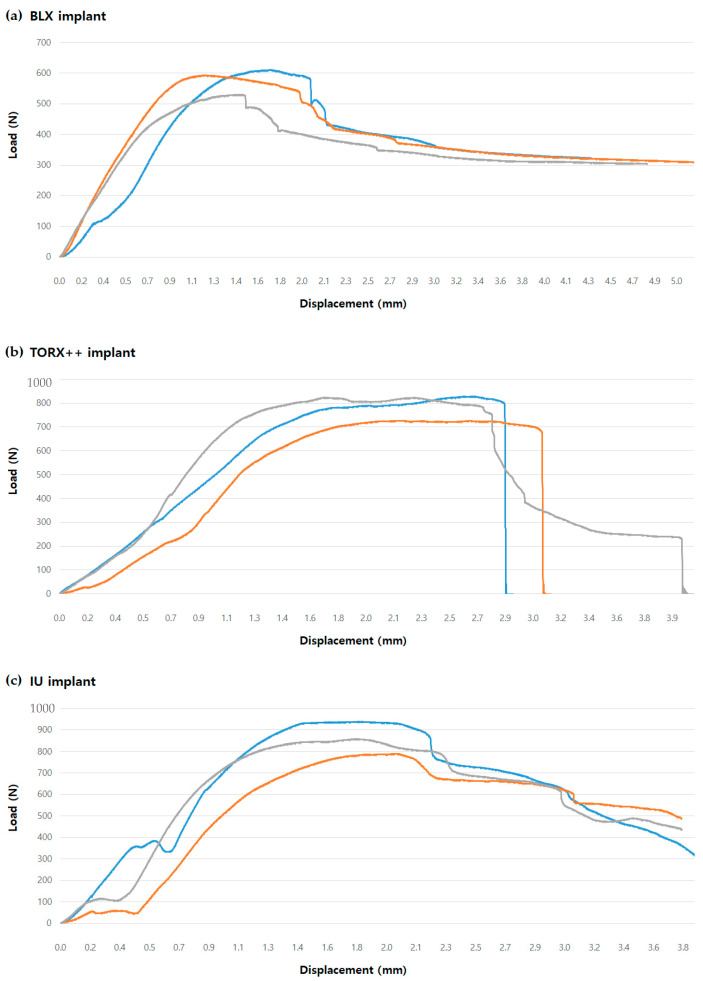
Load–displacement curves obtained in cyclic fatigue tests: (**a**) BLX implant; (**b**) TORX++ implant; (**c**) IU implant.

**Figure 3 jfb-13-00239-f003:**
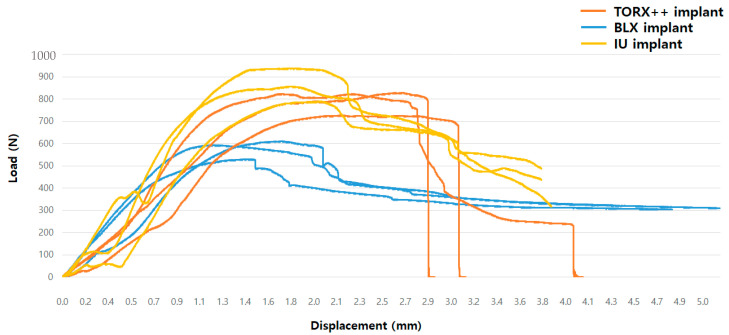
Load–displacement curves showing comparison among three groups in cyclic fatigue tests.

**Figure 4 jfb-13-00239-f004:**
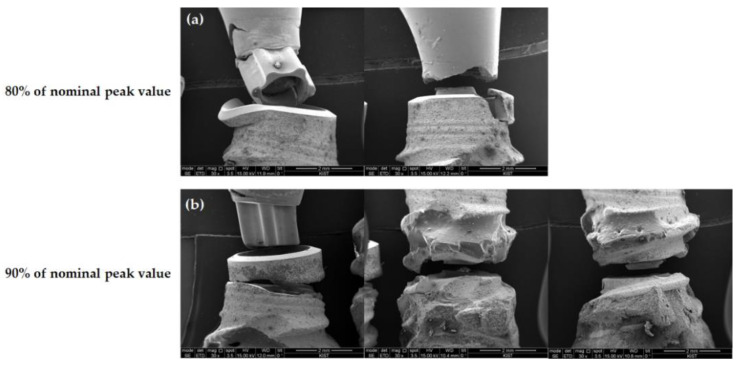
Front view of the fractured area in BLX samples observed by SEM (×30) (**a**) at 80% of nominal peak value and (**b**) at 90% of nominal peak value.

**Figure 5 jfb-13-00239-f005:**
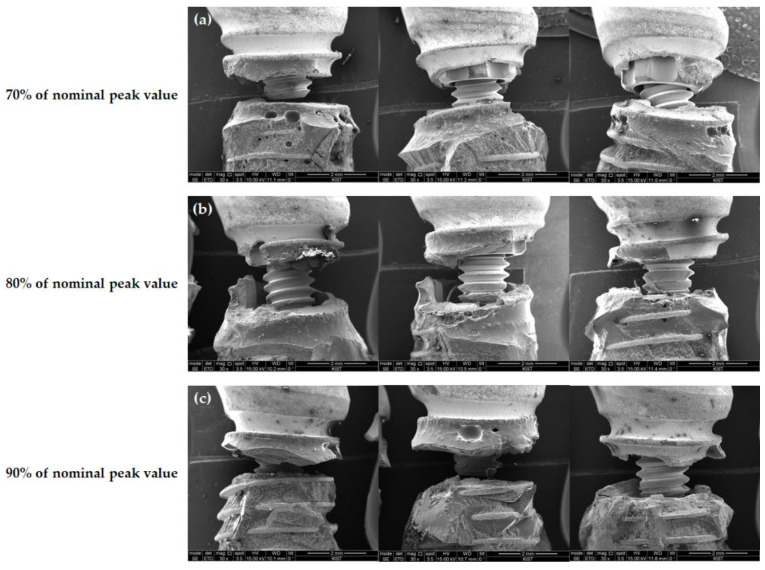
Front view of the fractured area in TORX++ samples observed by SEM (×30) (**a**) at 70% of the nominal peak value, (**b**) at 80% of the nominal peak value, and (**c**) at 90% of the nominal peak value.

**Figure 6 jfb-13-00239-f006:**
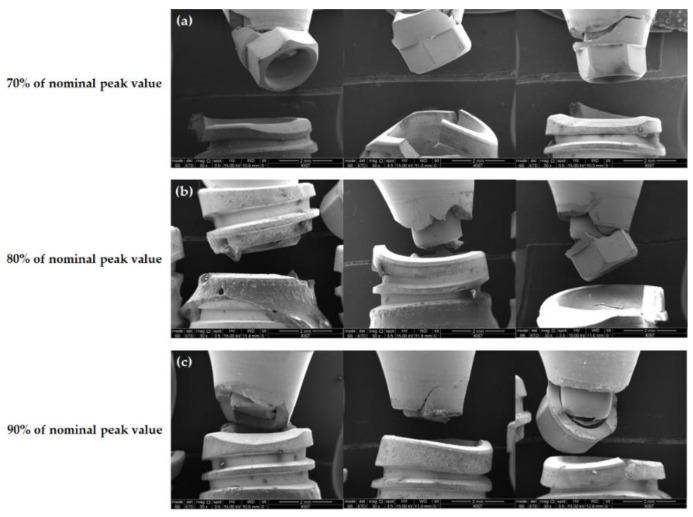
Front view of the fractured area in IU samples observed by SEM (×30) (**a**) at 70% of the nominal peak value, (**b**) at 80% of the nominal peak value, and (**c**) at 90% of the nominal peak value.

**Table 1 jfb-13-00239-t001:** Materials used in the present study. All fixtures and abutment components were made of commercially pure titanium (grade 4) except for the BLX implant fixtures, which were made of Ti–Zr alloy.

Components	Test Group 1 (*n* = 15)	Test Group 2 (*n* = 15)	Control Group (*n* = 15)
Fixture	TORX++ implant ^1^	IU implant ^2^	BLX implant ^3^
	(4.7 mm × 11 mm)	(4.5 mm × 10 mm)	(4.5 mm × 10 mm)
Abutment	Torx+ design	Internal hex design	Torx design
	Abutment height: 7 mm	Abutment height: 7 mm	Abutment height: 5.5 mm
	Gingival height: 2 mm	Gingival height: 6 mm	Gingival height: 3.5 mm

^1^ Manufacturer: Korea Dental Implant Inc., Ansan, Korea; ^2^ Manufacturer: Warantec Co., Ltd., Seongnam, Korea; ^3^ Manufacturer: Straumann, Basel, Switzerland.

**Table 2 jfb-13-00239-t002:** Result of the maximum breaking loads in single-load failure tests on specimens in each group.

	Load at Break (N)			
Products	First Specimen	Second Specimen	Third Specimen	Mean ± SD
BLX	611.22	593.66	530.48	578.45 ± 42.46
TORX++	828.68	727.00	824.11	793.26 ± 57.43
IU	938.71	790.43	857.74	862.30 ± 74.25

**Table 3 jfb-13-00239-t003:** Results of the fatigue tests according to the loading levels.

**BLX Implant**			
**Loading Level (%)**	**Sinusoidal Loading (N)**	**Number of Cycles Performed**	**Mean**
90	520	29,278; 78,008; 90,399	65,895
80	462	217,424; 501,002; 5,000,000	1,906,142
70	405 *	5,000,000; 5,000,000; 5,000,000	5,000,000
60	347	5,000,000; 5,000,000; 5,000,000	5,000,000
50	289	5,000,000; 5,000,000; 5,000,000	5,000,000
**TORX++ Implant**			
**Loading Level (%)**	**Sinusoidal Loading (N)**	**Number of Cycles Performed**	**Mean**
90	714	8745; 22,108; 6544	12,466
80	634	28,593; 42,415; 16,995	29,334
70	555	39,319; 25,529; 23,329	29,392
60	476	403,005; 227,229; 312,357	314,197
50	397 *	5,000,000; 5,000,000; 5,000,000	5,000,000
**IU Implant**			
**Loading Level (%)**	**Sinusoidal Loading (N)**	**Number of Cycles Performed**	**Mean**
90	776	219,622; 100,262; 8652	109,512
80	690	52,514; 34,935; 660,358	249,269
70	603	2,508,358; 114,156; 587,202	1,069,905
60	517	1,252,553; 5,000,000; 5,000,000	3,750,851
50	431 *	5,000,000; 5,000,000; 5,000,000	5,000,000

* Fatigue limit of each implant system.

## Data Availability

The datasets generated during and/or analyzed during the current study are available from the corresponding authors on reasonable request.

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
