# Peer review of "Fracture and Fatigue of Dental Implants Fixtures and Abutments with a Novel Internal Connection Design: An In Vitro Pilot Study Comparing Three Different Dental Implant Systems"

_jfb, 2022, doi:10.3390/jfb13040239_

Round 1
Reviewer 1 Report
Congratulation,
very interesting topic and research.
Nevertheless, some discussion should be made regarding the connection design (lenght and angulation) which can be also responsable for the results. This should be underlined in the conclussions as well, as a limitation of the study. Not only the materials or abutment heights are related with these types of failures, as was written in the article, BLT connection (from other similar studies) is a morse tapered connection that can dramatically change biomechanical behavior of roxolid implants.
Reviewer 2 Report
Thanks for the article and I have the following comments:
1. u-CT --> micro-CT or "mu"-CT ("mu" is the latin μ)
2. I appreciate Fig 1, but please indicate which sections from the micro-CT scan of the screw can get the 4 small sectioned images (respectively shown on the right of the screw)
3. Why n=3 was used in compress-to-fracture test? Do this test give fracture at the thinnest walls of implants? Is it a must to show individual data in Table 2 and Fig 2?
4. I do not understand, you mentioned n=12 for each implant, and how the 3 "number of cycles performed" values in Table 3 were obtained? and it looks like the SD are very large in the number of cycles. Can you explain why? I know you have been reported the mean values of these cycles but this looks like very odd because they are not comparable due to large SD...
5. Figs 3-5 seem illustrative, but there is no analysis for the fractography, nor analytically to calculate , say how may fractures are started at the thinnest point. The current presentation is not scientific enough.
Reviewer 3 Report
This manuscript is clinically interesting, but there are still some issues needed to be suitably clarified. The comments are listed in the following.
1. Are these three dental implant (fixture/abutment) systems used in this study commonly used in market?
2. Could the results obtained in this study be applied to other commercial dental implant (fixture/abutment) systems?
3. As recommended in ISO 14801, the frequencies for fatigue test of the dental implant system can be between 2 Hz and 15 Hz. Why did the authors choose the high frequency 15 Hz, instead of 2 Hz, for fatigue tests in this study? The frequency may have effect on the fatigue test results and 15 Hz seems to be far away from the real chewing frequency.
4. In Figure 2, the figure quality could be improved; the figure caption has typo ("cyclic loading tests"?).
5. P. 9, line 220: The authors cited the reference [16] which is related to the fracture resistance of “zirconia abutments”. Are the results of reference [16] suitable for discussion in this study? Please mention the abutment materials of the three implant systems used in this study, are they titanium, zirconia or both?
6. In References: most references are published before 2019 (only one reference in 2022). If possible, I would suggest to update some relevant references.
Reviewer 4 Report
The manuscript describes the mechanical behavior of three dental implant fixtures using static compression and dynamic fatigue studies. This manuscript shall be published after considering the following aspects.
In the title, it is provided as In vitro comparison study. The in vitro conditions used in this study needs to be described in the ‘Materials and Methods’ section.
Line 88 - ‘All mechanical tests were performed according to the ISO 14081:2016 protocol [8]’. Is this 14081 or 14801?
Line 88 - ISO 14801:2016 specifies a method of dynamic testing of single post endosseous dental implants of the transmucosal type in combination with their premanufactured prosthetic components. But the authors reported this standard has also been used for static uniaxial compression tests.
Line 105 - average 30° maximum compressive load – what does this imply? Is this the loading angle between the applied force and the axis of the implant?
It is suggested to explain how the sample was gripped during fatigue testing.
Line 119- From micro-CT analysis, the thinnest portions of each implant type is described. For the TORX++ implant it has been correlated in the fracture region during fatigue studies. It is suggested to mention whether the same can be applied for other two (BLX and IU) implant groups.
Line 127 – In Figure 1, it is recommended to insert a, b, c for the three implants for a better readability. Also in Figure 3,4, and 5, the same may be included and indicate in the caption what each image conveys.
For all the SEM images, scale bar should be incorporated.
Reviewer 5 Report
Comments on the paper:
The manuscript titled: "Fracture and Fatigue of Dental Implants Fixtures and Abutments with a Novel Internal Connection Design: An In Vitro Comparison Study of Three Different Dental Implant Systems" is very interesting article and worth of publication. The authors compared the mechanical behaviors of three dental implant fixtures with different abutment connection design and showed that all abutment connection designs display suitable mechanical properties for intraoral use. The static compression-to-fracture tests as well as fatigue tests were performed. Also, images of the implant-abutment assembles were obtained by micro-computed tomography and fractured or deformed areas were observed by SEM. The conclusions are consistent with the evidence and arguments presented in the paper and they address the main question posed. Also, tables and figures are given in informative manner.
However, it is not entirely clear whether all three implant systems are made of Titanium-Zirconium (Ti-Zr) alloy, since their comparison would make sense if they were made of the same material. Please, make it clear in the text.
Round 2
Reviewer 2 Report
Thanks and some amendments have been done. Just want to say:
1. "Why n=3 was used in compress-to-fracture test?" you can indeed calculate the sample size using suitable statistical parameters, or simply refer to any reference material such as journal article if you think you can achieve the statistical power. Your feedback "It would have been nice if more samples were used, but due to considerations of time and cost
for the study, it was decided to run with 3 samples" is not scientifically sound.
2. For load-cyclic fatigue test. OK to use ISO14801 that need at least 3 sample. However, you should do the load-cycle diagram , and also the "mean" cycles are not exist - I think the ISO 14801 needs to report the lowest peak load as the "maximum endured load" (LF) with the highest number of cycles (NF). Please change your entire results and rewrite the discussion.
3. If you cannot perform the fractography, please explain why. I guess you can analyse or at simply describe the fracture that you show here in Figs. 3-5 to let the readers what's happened. Simply mentioning "fracture" does not mean anything because the fracture can happen at any time at any load if you have a single void or crack according to Griffiths Law!
Round 3
Reviewer 2 Report
1. Thanks for pointing out the "n=3" issue for the static test. The article "Improvement in Fatigue Behavior of Dental Implant Fixtures by Changing Internal Connection Design: An In Vitro Pilot Study."
used n=3 so you have used the same for static-load fracture test. So, please insert the reference in line 70-72 at suitable place. In your line 267-268, the sentence "The limitations of this study include the absence of statistical comparison due to the relatively small sample size and the possibility of mixed results... " can change to "The limitations of this study include the absence of statistical comparison due to the relatively small sample size even the Choi et al. [ref no.] and ISO 14801 have used the same that might not be scientifically sound, and furthermore the possibility of mixed results ... "
On the other hand, you can also argue the large sample is for the precision of data only, not for bias (cf. https://www.sciencedirect.com/science/article/pii/S030057122200402X#bib0033 ) in mechanical test.
2. I had a look on Choi et al. paper, and their Table 4 exactly can be your results - just an analysis about the fracture mode is fine if you do not have a fractography data at the fractured surface.
I do not mind who clinician or engineer or scientist are writing the paper, but just to make it technically and scientifically sound. In our lab, all PhD with clinical background needs to learn science and engineering stuff even need to program the MATLAB / C++ / python by themselves. Those PhD with science/engineering background need to learn clinical terms too.
